# *Heortia vitessoides* Infests *Aquilaria sinensis*: A Systematic Review of Climate Drivers, Management Strategies, and Molecular Mechanisms

**DOI:** 10.3390/insects16070690

**Published:** 2025-07-02

**Authors:** Zongyu Yin, Yingying Chen, Huanrong Xue, Xiaofei Li, Baocai Li, Jiaming Liang, Yongjin Zhu, Keyu Long, Jinming Yang, Jiao Pang, Kaixiang Li, Shaoming Ye

**Affiliations:** 1Guangxi Colleges and Universities Key Laboratory for Cultivation and Utilization of Subtropical Forest Plantation, Guangxi Key Laboratory of Forest Ecology and Conservation, College of Forestry, Guangxi University, Nanning 530004, China; y2332856@163.com (Z.Y.); 18376763805@163.com (H.X.); lxx6337460@163.com (J.L.); 2Guangxi Key Laboratory of Special Non-Wood Forests Cultivation and Utilization, Guangxi Xylophyta Spices Research Center of Engineering Technology, Guangxi Forestry Research Institute, Nanning 530002, China; yychen2014@126.com (Y.C.); lixiaofeicaf@163.com (X.L.); treasurelii@163.com (B.L.); yongjin_z@126.com (Y.Z.); 3Beiliu Industrial Technology Research and Development Center, Yulin 537400, China; blgxcg@163.com (K.L.); blskjj@163.com (J.Y.); blqbs6221636@163.com (J.P.)

**Keywords:** *Heortia vitessoides*, *Aquilaria sinensis*, defensive genes, climate change, integrated pest control

## Abstract

In this paper, we review the effects of environmental factors on the development and biological characteristics of *Heortia vitessoides*, with emphasis on the genes associated with the environmental adaptability of both larvae and adults. Additionally, we examine the interaction mechanisms between *H. vitessoides* and host plants, and we summarize the current status of control measures. This work aims to provide new insights for the development of integrated control targeting *H. vitessoides*.

## 1. Introduction

*Heortia vitessoides* Moore (Lepidoptera: Pyralidae) is a pyralid moth whose leaf-chewing larvae severely damage the agarwood tree. In China, it is currently known to infest only *Aquilaria sinensis* (Lour.) Spreng. (Myrtales: Thymelaeaceae, commonly known as Chinese eaglewood) [1,2,3]. This species exhibits rapid developmental plasticity and high fecundity. Adults typically emerge at night and complete nocturnal mating and oviposition within a compressed 2–8 day lifespan [2,3,4,5,6,7]. Notably, density-dependent larval aggregation is mediated by cuticular hydrocarbon signaling, which optimizes resource utilization and collective survival [8,9].

The larvae of *H. vitessoides* primarily feed on the leaves of *A. sinensis*, exhibiting explosive population growth and voracious feeding behaviors. This pest can cause damage rates exceeding 90% in severely affected plants [4], with leaf damage rates reaching over 80% in heavily infested trees [10]. The population densities of this pest can range from hundreds to thousands per plant [2], leading to the rapid consumption of entire leaves and young branches within days. This extensive feeding activity can leave large areas of *A. sinensis* trees defoliated, potentially resulting in plant mortality in severe cases [11]. Research conducted by our project team in Guangxi revealed that *A. sinensis* plantations experience three to four significant outbreaks of *H. vitessoides* annually. Despite the trees’ strong recovery capability, severe infestations can result in the mortality of 10% to 30% of trees and significantly impede normal growth.

Geographically, *H. vitessoides* is distributed across tropical and subtropical regions, including southern China (Guangxi, Guangdong, Hainan), Southeast Asia, and parts of India and Australia [5,12,13]. Its population dynamics are tightly coupled with climatic factors such as temperature, humidity, and rainfall. Climate change is projected to expand its suitable habitats northward, potentially increasing invasion risks in previously unaffected areas like the Sichuan Basin by 2050 [14,15].

The pest’s expanding bioclimatic niche and host specialization create urgent challenges for agarwood production. Therefore, a comprehensive analysis of the occurrence patterns, pivotal developmental regulatory genes, and environmental adaptation mechanisms of *H. vitessoides* is imperative. Current management strategies for *H. vitessoides* predominantly focus on its biological characteristics, whereas studies on molecular adaptation mechanisms (e.g., detoxification genes, thermal stress responses) and host plant defense systems (e.g., JA signaling pathways) remain fragmented. This compartmentalization hinders the development of holistic pest control frameworks. To bridge this gap, our review adopts a multi-scale perspective, integrating three critical dimensions: (1) the ecological drivers of *H. vitessoides* population dynamics (e.g., climate-driven developmental plasticity), (2) molecular adaptations underpinning its environmental resilience, and (3) tritrophic interactions among the pest, its host *A. sinensis*, and natural enemies. By synthesizing these domains, we aim to identify synergies between genetic insights and field-based control technologies, ultimately proposing a precision management framework that aligns molecular targets with ecological regulation.

## 2. Biological Characteristics, Occurrence Patterns, and Climate Response of *Heortia vitessoides*

### 2.1. Growth and Development Characteristics

The eggs of *H. vitessoides* are oblate, densely arranged in scales, and measure 0.5 to 0.8 mm in diameter (Figure 1A). Hatching occurs within 3 to 4 days, with a high rate exceeding 95% (laboratory data) [5]. The majority of eggs hatch into larvae between 22:00 and 06:00 the following day [3]. The larvae undergo five instars based on the number of molting events during post-hatching growth [5]. These instars span specific age periods: 1st to 4th day, 4th to 6th day, 6th to 10th day, 9th to 12th day, and 11th to 16th day. Notably, *H. vitessoides* exhibits aggregation and feeding behaviors, and its growth and development are influenced by group size. Larvae reared in isolation die within two days without evidence of leaf consumption [16], while both newly hatched and second-instar larvae display non-kin aggregation tendencies [16]. From the third instar onward, large groups (90 larvae) exhibit significantly greater body length and developmental rates than small groups (30 larvae), despite comparable mortality across developmental stages [8]. Chemical analyses reveal that hexane extracts of second-instar larval cuticular chemicals induce conspecific aggregation, an effect absent in fifth-instar extracts [9]. Conversely, acetone extracts from both instars repel second-instar larvae [9]. These findings suggest leveraging aggregation behavior for biocontrol, e.g., abamectin treatment reduces survival and adult emergence through horizontal transfer [16]. Additionally, the species-specific attractant and repellent effects of larval cuticular chemicals provide novel strategies for integrated pest management. Upon reaching the mature fifth-instar stage, the larvae burrow 1 to 2 inches deep into the soil to spin a cocoon and pupate within approximately 2 days [5,17,18]. Study findings indicate that substrates with 20% to 60% humidity are conducive to the pupation of *H. vitessoides* [19,20]. The pupal stage lasts 8 to 15 days [5,18]. Adult moths show sexual dimorphism in abdominal color and structure (Figure 1E,F). Adult moths can emerge at any time during the day, with peak emergence occurring between 20:00 and 22:00. After emergence, the adults engage in activities such as feeding on nectar, mating, and oviposition, displaying strong phototaxis. Their lifespan ranges from 2 to 8 days [2,3,4,5,6,7].

### 2.2. Occurrence Patterns of Heortia vitessoides in Different Regions

*H. vitessoides* is prevalent in major agarwood-producing regions, including China and Southeast Asia, and it has been documented in India (Assam, Meghalaya, Nagaland, Sikkim, Tamil Nadu, and Tripura), Nepal, Bangladesh, Sri Lanka, Australia, Fiji, and other tropical and subtropical regions [5,12,13]. The frequency and timing of *H. vitessoides* outbreaks vary by region (Table 1). Regional population phenology exhibits latitudinal gradients. In China, *H. vitessoides* is primarily found in Guangdong, Guangxi, Hainan, Yunnan, and Fujian. In Hainan, it undergoes 8 to 10 generations annually, with pest activity from February to November, peaking in April [4,21]. In Guangdong, six to eight generations occur yearly, with infestations from April to December. Each generation lasts about one month, with the second generation showing some overlap. The final generation of larvae begins pupation and overwintering in mid-December [5]. In Guangxi, five to six generations can occur annually, with the peak period concentrated between April and November. Last-instar larvae typically enter the soil for pupation and overwintering in late November. Adult emergence occurs in early March of the following year, with the first peak of damage occurring from late April to early May [22]. Some studies suggest that damage can also occur in mid-to-late December, possibly influenced by factors such as temperature, humidity, and rainfall [22]. Reports indicate that *H. vitessoides* exhibits six generations annually in Yunnan, with peak damage occurring from April to May, reaching a damage rate of up to 90% [23]. In India, this pest undergoes four to five generations per year, with activity observed from February to September and a life cycle duration of approximately 32 to 35 days [24]. In Indonesia, *H. vitessoides* is present year-round, causing the most significant damage from July to September during the dry season [12]. With global climate change, there may be a significant migration risk in the distribution area of *H. vitessoides*. Climate-driven distribution models project northward range expansion, identifying current high-suitability zones in tropical and south subtropical river basins, with the Sichuan Basin emerging as a new risk area by 2050, influenced by changes in precipitation and temperature [14,15]. These bioclimatic shifts necessitate urgent investigation into thermal-humidity thresholds governing pest–host synchrony and co-evolutionary dynamics under climate change scenarios. It is essential to investigate the influence of climate factors on the changes in the suitable habitats of *H. vitessoides* and its host *A. sinensis*.

### 2.3. Climate Response of Heortia vitessoides

Temperature, humidity, and rainfall are key factors influencing the growth, development, and extent of damage of *H. vitessoides* [4,12,14,15,17,23,25]. *H. vitessoides* can complete generational development at 15–31 °C, with developmental stages and generational periods shortening as temperatures rise. However, extreme temperatures, whether too high or too low, are detrimental to growth and development. Research indicates that larval developmental abnormalities can occur at 11 °C, while eggs fail to hatch at 35 °C [23]. Zhou et al. [25] proposed that the monthly dynamic variation of *H. vitessoides* in Hainan is primarily influenced by temperature fluctuations. By applying the effective accumulated temperature (EAT) model [26] [The effective accumulated temperature model comprises two parameters: developmental zero temperature (*T*_0_) and EAT (*K*). Its core principle states that the rate of development of an insect is directly proportional to the temperature, allowing for predicting phenological changes in agricultural pest life cycles.], it was estimated that *H. vitessoides* in Hainan undergoes 8–10 generations annually, aligning closely with investigation data. Several studies have demonstrated that *H. vitessoides* infestations are more prevalent during the dry season, with sustained high temperatures and rainfall hindering *H. vitessoides* growth, development, and activities [12,14,15,17,23]. Heavy rainfall directly limits larval feeding and hinders pupal development by increasing soil moisture, which promotes Aspergillus fungal infections. When soil moisture exceeds 80%, only a few last-instar larvae successfully complete the pupal stage [12,17]. Additionally, rainfall impedes adult mating and oviposition activities. *H. vitessoides* adults rarely oviposit during rainy conditions, but oviposition peaks 1–2 days post-rain [12,23], possibly due to rain disrupting adult sex pheromone recognition. European corn borer (*Ostrinia nubilalis*) studies indicate that increased humidity (43–100%) reduces male sensitivity to sex pheromones [27]. Furthermore, low temperatures may delay male moth responses to sex pheromones [28], though research on this aspect of *H. vitessoides* is limited. Therefore, further investigation into the effects of climate on sex pheromone production and perception in *H. vitessoides* is warranted.

Beyond affecting developmental processes across insect life stages, climate change also influences insect population dynamics through multiple direct and indirect pathways. These mechanisms often involve complex interactions with factors such as host plant resource availability, predator abundance, and seasonality. Regarding direct effects, rising temperatures may exceed physiological tolerance thresholds, increasing risks of heat stress or cold exposure. For example, in the spruce budworm (*Choristoneura fumiferana*), host plant (*Abies balsamea*) defoliation induced an 18–22 day phenological mismatch between larvae and leaf development at 12 °C. Elevating the temperature to 22 °C reduced this mismatch to 3–7 days. This enhanced synchrony improved food quality, promoting larval development and indirectly increasing outbreak risk by narrowing the developmental time gap between the insect and its host plant [29]. Similarly, in *Spodoptera exigua*, the total developmental duration shortened to 17.86 days at 35 °C. However, extreme high temperatures (40 °C) completely suppressed egg hatching, demonstrating the “dual effect” of temperature on population growth [30]. Furthermore, although warmer winters reduce the risk of extreme cold events, they may lead to premature depletion of energy reserves in overwintering insects, potentially triggering population collapse [31]. Regarding indirect effects, climate change reconfigures species interaction networks by altering resource quality and natural enemy dynamics. Elevated temperatures modulate the synthesis of plant secondary metabolites (e.g., monoterpenes), affecting nutrient acquisition by phytophagous insects. For instance, temperature impacts on pine sawflies (*Neodiprion sertifer*) exemplify tritrophic plant–insect–enemy interactions. Increased diterpene concentrations in pines (*Pinus sylvestris* L.) under high temperatures directly suppress pine sawfly survival. However, pine sawflies incorporate ingested diterpenes into their defense systems (e.g., via internal storage or cuticular deposition), conferring protection against higher-trophic-level predators or parasitoids. This biochemical co-option indirectly enhances survival, altering interspecific balances. Concurrently, elevated CO_2_ concentrations may suppress the jasmonic acid (JA) pathway in plants, reducing chemical defenses against chewing insects while diminishing volatile organic compound (VOC) emissions, thereby compromising parasitoid host-location efficiency [32,33]. Furthermore, heterogeneity in host plant nutritional composition modulates insect fitness. For instance, the higher protein content in soybean leaves (47.54%) compared to maize (29.78%) enhances larval development in the beet armyworm (*Spodoptera exigua*). However, this nutritional advantage attenuates under elevated temperatures due to accelerated metabolic rates, resulting in diminished inter-host developmental differentials [30]. Seasonal phenological mismatch between insects and host plants represents a quintessential climate-driven indirect effect. Its severity depends on host plant resource abundance and insect population density compensation mechanisms. In the winter moth (*Operophtera brumata*), spring warming caused larval hatching to precede oak budburst. Following a decade of natural selection, the population underwent adaptive evolution by delaying hatching time, thereby mitigating resource asynchrony. However, density-dependent compensation can buffer the negative impacts of mismatch. After partial mortality from asynchrony, reduced competition enhances fecundity in surviving individuals, maintaining population stability [31]. Seasonal variations further modulate population fluctuations by altering predator-prey dynamics. For instance, elevated temperatures may increase predation rates by enhancing predator metabolism (e.g., heightened predation by dyctisidae beetles on mosquito larvae under warming). Conversely, rising temperatures can disrupt chemical signaling in predators. In addition, beetle larvae release more pheromones at higher temperatures, leading to reduced oviposition by conspecific females. The interplay of these stimulatory (enhanced predation) and inhibitory (reproductive suppression) effects generates complex nonlinear regulatory networks [33]. These multi-scale interactions underscore the complexity of insect population responses under climate warming, necessitating joint analysis of ecological and evolutionary consequences through long-term monitoring and multi-trophic experiments.

While the impact mechanisms of climate change on the population dynamics of most insect species have been extensively studied, research on the multidimensional regulatory mechanisms by which climate change influences the population dynamics of *H. vitessoides* remains limited. Furthermore, developing green control technologies based on these climatic response mechanisms requires balancing environmental adaptability with ecological safety. The following sections will systematically evaluate the potential and limitations of existing control techniques, including physical trapping, chemical control, and natural enemy utilization. Additionally, the role of population dynamics modeling in optimizing control strategies will be explored.

## 3. Management Strategies for *Heortia vitessoides*

### 3.1. Physical Trapping

Compared to other control methods, light trapping offers the advantages of avoiding environmental pollution and mitigating issues related to pest resistance. A previous study found that the trapping efficiency of frequency vibration insecticidal lamps significantly surpassed that of blacklight and fluorescent lamps. A single lamp could control insects in an area of approximately 3.14 hectares, with an optimal hanging height of 2 m above the ground [6]. By combining *Kuhnia rosmarnifolia* and *Santalum album*—plants that attract adult *H. vitessoides*—with frequency killing lamps, each lamp has the capacity to capture hundreds to thousands of *H. vitessoides* adults daily [34], but installation costs and reliance on electricity are limiting factors. Moreover, insecticidal lamps may attract and kill non-target insects, disrupting ecological balance. Studies indicate a significant 14% decline in macro-moth abundance within the study area during the final three years of a five-year continuous artificial lighting regime [35]. Furthermore, during high-density outbreak periods, uncaptured males can mate multiply with females, limiting offspring population suppression. Therefore, this control method is unsuitable for ecologically sensitive areas and pest outbreak phases. Future refinements should focus on *H. vitessoides*-specific pheromone trapping or mating disruption—releasing high-concentration sex pheromones to interfere with olfactory cues and prevent mating [36].

### 3.2. Chemical Control

The insecticidal efficacy of chemical agents against *H. vitessoides* has been extensively studied (Table 2). A 5.0 × 10^6^ dilution of 1.8% avermectins EC and a 5.0 × 10^6^ dilution of 0.5% emamectin benzoate microemulsion achieved 100% mortality against fifth-instar larvae in forest settings [37]. Similarly, a 30-fold dilution of Sendebao (a mixture of avermectin B1a and *Bacillus thuringiensis*) and an 8000-fold dilution of 3% fenoxycarb exhibited a corrected mortality rate of 98.9% against mature larvae [38]. For mild infestations, a 1000-fold dilution of 5% eucalyptol SL showed a significant increase in corrected mortality (90.88%) two days post-treatment [39]. Additionally, spinetoram and spinetoram·methoxyfenozide, applied at 1000-fold dilutions, resulted in 100% lethality against *H. vitessoides* larvae in laboratory assays [6]. Biological agents have also demonstrated high efficacy (Table 2). *Helicoverpa armigera* NPV (2 × 10^9^ PIB/mL) and 0.3% Matrine showed significant lethality and low LC50 values against all five instars of *H. vitessoides* [40]. Entomopathogenic fungi, such as *Beauveria bassiana* and *Metarhizium anisopliae*, have been effective in field experiments (Table 2). Concentrations of 2.4 × 10^8^ and 2.4 × 10^10^ spores/mL of *B. bassiana* and *M. anisopliae* induced mortality rates of 84–90% and 40–52%, respectively, in *H. vitessoides* larvae [24]. Notably, a high concentration of *M. anisopliae* (1 × 10^9^ spores/mL) achieved 100% mortality. Furthermore, combining sublethal concentrations of *M. anisopliae* with chemical insecticides demonstrated a synergistic effect, significantly enhancing larval mortality and offering a novel approach to reducing chemical pesticide usage [41]. In a recent discovery, Zhao et al. [42] isolated *Aspergillus nomius* Q527 from deceased *H. vitessoides* larvae, marking the first reported association between this fungus and *H. vitessoides*. Inoculation with *A. nomius* Q527 resulted in an 83.3% mortality rate in larvae seven days post-treatment, highlighting its potential as a biocontrol agent. However, the release of biological agents such as pathogenic bacteria may infect non-target insects and soil microbes, with potential spore dispersal [43]. For instance, studies show that applications of *Bacillus thuringiensis* targeting mosquito larvae significantly reduce the abundance of chironomids and crustaceans (meta-analysis indicates ~50% decline in chironomids). Spores can disperse onto plant foliage via rain splash, remaining viable for extended periods—half survived at 120 days post-application, with 20% persisting after one year [44,45]. Furthermore, synergistic application with chemical pesticides may increase environmental residues, impairing ecosystem functions. Future research must rigorously evaluate their long-term ecological impacts.

### 3.3. Prevention and Control of Natural Enemies

Various natural enemy insects have demonstrated effective control against *H. vitessoides*. Parasitic insects, including *Trichogramma pintoi*, *T. ostriniae*, *T. chilonis*, and *T. dendrolimi*, can parasitize the eggs of *H. vitessoides*. The parasitization rate of *Trichogramma chilonis* on egg masses was the highest, reaching 85.34% [46,47,48]. Furthermore, *Trichogramma evanescens* exhibited an average parasitization rate of 55.74% in forest environments. When released twice—initially and four days after oviposition—the average oviposition parasitization rate increased to 81.03%, effectively reducing larval population density [49,50,51]. Additionally, predatory insects such as *Eocanthecona furcellata* and *Hierodula petellifera* have been observed preying on *H. vitessoides* larvae [52,53,54]. It is noteworthy that while biological control strategies utilizing native natural enemies generally pose lower ecological risks, generalist natural enemies may impact beneficial native insects through direct predation or indirect resource competition. This can lead to niche competition, potentially compromising biological control efficacy and even disrupting ecological balance [55,56]. Therefore, future efforts for *H. vitessoides* control must incorporate species-specific assessments (e.g., dispersal capacity, host range, niche overlap) to ensure ecological safety.

### 3.4. Modeling of the Population Dynamics of Heortia vitessoides

Population dynamic modeling provides the basis for predicting pest outbreaks and evaluating control efficacy, thereby facilitating precision and ecological approaches to *H. vitessoides* control to achieve green pest management. Xu et al. [15] utilized species distribution data (237 global occurrence records of *H. vitessoides* from field surveys, CABI, GBIF, NSII, China Animal Scientific Database, and National Animal Collection Research Center), and environmental variables (19 bioclimatic variables from WorldClim for current (1950–2000) and future (2050s) scenarios from CCAFS for MaxEnt modeling. Host plant distribution (*A. sinensis* from GBIF and CVH) was compared post-modeling to explain range expansion trends. Variables were filtered using the Jackknife test (contribution > 1% and correlation coefficient < 0.8), and eight key predictors were retained, including precipitation of wettest/warmest/driest quarters, isothermality, mean diurnal temperature range, precipitation seasonality, mean temperature of warmest quarter, and minimum temperature of coldest month. The final model demonstrated high predictive accuracy (current AUC: 0.981; future scenarios AUC: >0.97), exceeding the excellent threshold (AUC > 0.9), and is suitable for predicting the pest’s potential spread risk in China. Notably, integrating climate adaptation-related gene research can significantly enhance the accuracy of ecological models. The core lies in combining species’ molecular adaptation mechanisms with environmental variables, thereby enabling more precise prediction of population responses to climate change. For instance, studies on *H. vitessoides* have identified key genes (e.g., *HvCAT*, *HvGP*) strongly associated with thermal adaptation. Their expression patterns not only reveal physiological response thresholds to different temperatures but also serve as biomarkers quantifying the population’s climate adaptation potential. Integrating these adaptive traits into models addresses limitations of traditional species distribution models (SDMs, such as MaxEnt), which rely solely on environmental variables (e.g., temperature, salinity). For example, gradient forest models [57] can predict genomic offset under future climates by analyzing genotype–environment associations, thereby assessing the match between adaptive requirements and habitat suitability. Similarly, future research on *H. vitessoides* should validate gene functions using techniques like CRISPR/Cas9 and incorporate multi-gene synergy into model parameters to further enhance the reliability and practical utility of climate adaptation predictions.

Furthermore, dynamic modeling can assess the efficacy of different pest control strategies. Existing studies typically integrate mathematical models based on pest biology and control measure mechanisms, combined with threshold analysis and parameter sensitivity assessment, to optimize decision making. Furthermore, Anguelov et al.’s mating disruption model, formulated using a piecewise-smooth ODE system, incorporates pheromone trapping efficiency (α) and dosage (Y_p_) as key parameters. Theoretical analysis identified two critical thresholds: the initial effective threshold (Y_p_*) and the extinction threshold (Y_p_**). Numerical simulations demonstrated that eradication required only Y_p_ = 102,462 for an initial population of 1000 individuals, versus Y_p_ = 987,735 for a traditional no-trapping scenario. This parametric comparison provides a quantitative tool for cost-benefit analysis [58]. Similarly, Xiang et al.’s impulsive differential equation model, based on a pest–predator–pathogen coupling system, derived the threshold condition R_0_ for pest eradication. The model revealed a non-linear relationship between spray frequency and release period. When insecticide lethality against infected pests was low (*p* = 0.2), optimal control occurred at 2–3 sprays per period, whereas higher lethality (*p* = 0.35) reduced efficiency with increased spraying. This quantitative analysis offers a theoretical basis for determining optimal intervention windows [59].

Dynamic modeling integrates parameters for pest development, environmental response, and control measures. Utilizing threshold analysis, sensitivity testing, and scenario simulation, it systematically evaluates the ecological effects and economic costs of different strategies to identify optimal intervention combinations. Its core value lies in revealing non-linear interactions, predicting long-term dynamics, and avoiding the high costs and ecological risks of field trials. Current research on population dynamic modeling for *H. vitessoides* control is limited. Future work should develop delay differential equation models incorporating larval developmental stages, adult dispersal capacity, and the regulatory effects of various control measures. Simulating population dynamics under different control scenarios would enable comparison of control costs and efficacy. Integrating MaxEnt models for species distribution prediction could ultimately determine regionally adaptive green control strategies. Furthermore, while population models predict pest distribution and outbreak trends, integrating molecular mechanism research is needed to enhance precision. Key drivers of *H. vitessoides* environmental adaptation include gene networks governing detoxification, thermotolerance, and chemosensation. This section will elucidate the mechanisms of key functional genes and reveal multidimensional regulatory pathways from genes to environmental adaptation.

## 4. Molecular Mechanism of *Heortia vitessoides*: From Gene to Environmental Adaptation

In the last decade, research on the functional genes in *H. vitessoides* has garnered increasing attention, with reported genes primarily associated with insect detoxification, temperature stress, metamorphosis, and chemical perception. Law et al. [60] initially described a 517 Mb genome with 2n = 62 chromosomes, including Z and W sex chromosomes and 30 pairs of autosomes. They identified 635 expanded gene families, particularly involved in detoxification, metabolism, and growth and development, such as C-type lectins, cytochrome P450, and UDP-glucuronosyltransferases, providing a foundational dataset for future investigations into the functional genes of *H. vitessoides*. To comprehensively review the current research on the functional genes of *H. vitessoides*, we synthesized information on subcellular localization, expression patterns, regulatory mechanisms, and functions, as outlined in Table 3.

### 4.1. Toxin-Related Genes

The detoxification enzyme system is a crucial adaptation in insects for plant feeding, playing a vital role in breaking down foreign toxins and maintaining normal physiological metabolism. Insects can reduce the efficacy of insecticides through mechanisms such as decreased penetration, enhanced sequestration, or increased detoxification. This results in decreased target sensitivity or target site modification, rendering the insecticide ineffective. This process involves three key protein families responsible for pesticide metabolism: cytochrome P450, carboxylesterases (COEs), and glutathione S-transferases (GSTs) [61]. GSTs represent a highly diverse gene family [62,63,64]. In insects, GSTs are classified into six subclasses: Delta, Epsilon, Sigma, Omega, Theta, and Zeta [65]. To date, the reported GST gene in *H. vitessoides* belongs to the Sigma class and is designated as *HvGSTs1*. In a previous study, the expression of *HvGSTs1* showed significant variation across different developmental stages and tissue types in *H. vitessoides*. Notably, *HvGSTs1* expression was significantly higher in the pupal stage than in the egg, larval, and adult stages, suggesting a potential association with metabolic dormancy. Furthermore, in fourth-instar larvae, *HvGSTs1* expression was significantly higher in the fat body than in the midgut, head, and tegument. In adults, *HvGSTs1* expression levels were significantly higher in the abdomen and thorax than in the head, legs, and wings. These findings collectively suggest that *HvGSTs1* is involved in the emergence and detoxification processes of *H. vitessoides*, particularly in the larval fat body. Additionally, *HvGSTs1* may play a role in detoxifying endogenous toxic substances that accumulate in the abdomen and thorax of adults [66] (Table 3). Moreover, studies have shown that COEs play a significant role in the detoxification of *H. vitessoides*. When larvae were fed on highly resistant versus susceptible plants, GST activity in fifth-instar larvae showed no significant change. However, significant variations were observed in COE and acetylcholinesterase (AChE) activities. Larvae fed on resistant *A. sinensis* plants showed significantly reduced food utilization and approximate digestibility, accompanied by increased COE activity and inhibited AChE activity [67]. These findings highlight the complexity of the detoxification process in *H. vitessoides*, underscoring the need for further investigation into various detoxifying enzyme families such as COEs and GSTs to elucidate the underlying mechanisms.

### 4.2. Genes Related to Temperature Stress

Temperature is a key climatic factor influencing *H. vitessoides* outbreaks. Temperature changes directly affect insect growth and development and also impact their sensitivity to biotic (e.g., natural enemies, pathogens) and abiotic (e.g., insecticides) stress factors [68,69]. Understanding the mechanism of *H. vitessoides* in response to temperature stress is crucial for its pest monitoring, early warning, and pest management. Antioxidation and energy metabolism are two vitally important strategies that insects employ to cope with temperature stress [70]. Adverse conditions, such as extreme temperatures, lead to a rapid increase in reactive oxygen species (ROS) levels in insects, causing oxidative damage. Insects mitigate ROS accumulation by producing various antioxidant enzymes, such as catalase (CAT), peroxidase (POD), glutathione S-transferases (GSTs), thioredoxin peroxidase (TPX), and superoxide dismutase (SOD) [71,72,73,74]. A previous study found that, under temperature stresses of 0 °C, 10 °C, and 35 °C, the expression of the *HvTpx* gene in fourth-instar larvae of *H. vitessoides* was significantly elevated compared to that of the control at 25 °C. Conversely, at extreme temperatures of −15 °C and 40 °C, *HvTpx* expression was markedly downregulated, suggesting that extreme low or high temperatures exceed the tolerance range of TPX [75]. Similarly, both the transcriptional expression and enzyme activity of *HvCAT* increased significantly at 35 °C, with larval mortality remaining below 20%. However, the RNAi-mediated silencing of *HvCAT* in fifth-instar larvae accelerated the death of larvae and resulted in over 80% mortality (at 35 °C), underscoring *HvCAT*’s critical role in high-temperature resistance in *H. vitessoides* (Table 3). Additionally, *HvTpx* expression significantly increased 48 h post-*HvCAT* silencing, possibly due to a compensatory upregulation of expression [76]. Beyond antioxidant stress responses, *H. vitessoides* adapts to extreme temperatures by modulating energy metabolism-related proteins. Studies have shown that glycogen phosphorylase (GP) is involved in glycogen degradation and helps insects resist low temperatures [77,78,79]. GP is highly expressed in the fat body of fourth- to fifth-instar *H. vitessoides* larvae, contributing to rapid temperature stress responses. Exposure to low temperatures (5–20 °C) for 90 min significantly upregulates *HvGP* expression, with peak expression observed at 10 °C. Conversely, gene expression is notably downregulated at 0 °C and under thermal stresses at 30–40 °C. This suggests that GP activation by cold stress facilitates the conversion of glycogen into antifreeze protectants in *H. vitessoides*, although temperatures below 0 °C or above 30 °C may exceed GP’s effective range [80]. Arginine kinase (AK) is ubiquitously found in invertebrates and plays a crucial role in energy metabolism [81]. In contrast to *HvGP*, *HvAK* exhibits significantly elevated expression levels at extreme temperatures compared to normal temperatures. Notably, when the temperature exceeds 30 °C, *HvAK* expression remains upregulated in fourth-instar *H. vitessoides*, with even more pronounced upregulation observed under 4 °C conditions [82]. These findings suggest that *HvAK* and *HvTpx* may regulate the adaptability of *H. vitessoides* to extreme temperatures on a larger scale, potentially indicating compensatory interactions between different protein systems. Therefore, a comprehensive exploration of diverse enzyme systems (such as SOD and POD in the antioxidant system) in response to temperature fluctuations is essential. This will help unravel the mechanisms of temperature adaptation in *H. vitessoides*, a key future research direction.

### 4.3. Genes for Metamorphosis of Heortia vitessoides

Molting is a crucial aspect of insect metamorphosis and growth, involving the regeneration of a new cuticle by synthesizing chitin and other compounds from epidermal cells, followed by the shedding of the old exoskeleton [83,84]. The main reported molt-related proteins include trehalose-6-phosphate synthase (TPS), chitin deacetylase (CDA), ecdysone receptor (EcR), and fatty acid binding protein (FABP). Insects can produce trehalose with the assistance of TPS [85]. Trehalose not only aids in the regulation of various biotic and abiotic stress responses [86] but also contributes to the control of chitin synthesis and degradation during insect molting [85,87]. Chen [88] employed RNA interference (RNAi) to disrupt *HvTPS* expression, which led to the inhibition of trehalose synthesis. Trehalose serves as the precursor for chitin biosynthesis. Consequently, ds*HvTPS* treatment downregulated the expression of pivotal genes in the chitin biosynthetic pathway, impeding chitin production in the larval cuticle and midgut, thereby delaying the developmental period of *H. vitessoides*. Subsequently, the survival rate of fourth-instar larvae notably declined during the ensuing developmental stages, with only a 23.00% success rate in emergence, resulting in adults with malformed wings. Anatomical examinations revealed a marked increase in fat body size due to *HvTPS* silencing. Moreover, the expression levels of genes associated with fatty acid synthesis, namely, *HvACC* (acetyl-CoA carboxylase) and *HvFAS* (fatty acid synthase), were significantly upregulated, while the expression of *HvLIP1* (lipase 1), involved in fatty acid breakdown, decreased substantially. These findings suggest that *HvTPS* plays a significant role in regulating both chitin and lipid biosynthesis. The chitin deacetylase genes *HvCDA1* and *HvCDA2* are crucial for the metamorphosis of *H. vitessoides*. Among the cuticle, head, fat body, foregut, midgut, and Malpighian tubules of fifth-instar larvae, the expression levels of *HvCDA1* and *HvCDA2* were the highest in the cuticle, increasing before or after molting. In different developmental stages, the expression levels of the two genes were significantly higher in the pupal stage than in the larval stage, and the silencing of these genes led to abnormalities and even death during molting, pupation, or emergence [89]. EcR is crucial for insect molting and metamorphosis. *HvEcR* expression was notably higher in fifth-instar larvae than in other developmental stages, indicating its significance in the development of last-instar larvae. *HvEcR* expression was highest in the fat body of fourth- and fifth-instar larvae compared to that in the cuticle, head, midgut, and Malpighian tubules. The fat body serves as a key energy reservoir in insects, essential for metabolism, suggesting a potential role of *HvEcR* in energy storage and metabolism in *H. vitessoides* [90]. Additionally, the fatty acid binding protein gene (*HvFABP*) is crucial for molting, as silencing this gene results in molting failure and subsequent death in *H. vitessoides* [91].

### 4.4. Genes Related to the Chemoreceptor of Heortia vitessoides

Insects’ antennae serve as crucial sensory organs containing various types of chemical receptors that enable insects to detect plant odors, chemical pheromones, sex pheromones, and volatile compounds emitted by both host and non-host plants. These chemical signals play a vital role in various insect activities such as mating, feeding, oviposition, host identification, predator avoidance, and other life-sustaining behaviors [92]. The mouthparts represent the primary feeding organ in insects, housing taste receptors [93]. A total of 124 chemoreceptor genes have been identified in the antennae and mouthparts of the adult: 50 odorant receptor (OR) genes, 19 ionotropic receptor (IR) genes, 17 gustatory receptor (GR) genes, 19 odorant-binding protein (OBP) genes, 17 chemosensory protein (CSP) genes, and 2 sensory neuron membrane protein (SNMP) genes. It is noteworthy that three sex pheromone receptors (PRs) were identified among 50 ORs in *H. vitessoides*. *HvitOR42* and *HvitOR43* exhibited high expression levels in male antennae, surpassing female expression by more than 10-fold, suggesting their potential role in female sex pheromone recognition. Conversely, *HvitOR20* showed significantly higher expression in female antennae (more than 10 times higher than in males), indicating its probable involvement in female oviposition behavior. Gender-specific expression patterns were also observed in OBPs and CSPs. Among the mouthparts, eight OBPs, including *HvitGOBP2*, *HvitPBP1*, and *HvitPBP2*, displayed markedly higher expression in males, exceeding female expression by more than 10-fold. Additionally, *HvitCSP8*, *HvitCSP15*, and *HvitCSP17* were preferentially expressed in males, with expression levels more than three times higher than in females, potentially associated with sex pheromone recognition [94].

### 4.5. RNAi Technology: Bridging Genomic Research and Pest Control in Heortia vitessoides

Current genetic research in pest control primarily focuses on gene drives, RNA interference (RNAi), CRISPR-Cas9 gene editing, the sterile insect technique (SIT), and symbiont manipulation (e.g., Wolbachia) [95,96,97,98,99]. Current *H. vitessoides* genetic research employs RNA interference (RNAi) solely for gene functional studies, with no practical control applications yet implemented. RNAi-based pest management primarily utilizes two approaches: (1) transgenic plant-mediated RNAi, and (2) exogenous dsRNA formulations [100]. Transgenic plant-mediated RNAi involves engineering host plants to continuously express dsRNA targeting essential pest genes. For example, SmartStax^®^ Pro corn expresses dsRNA targeting the *DvSnf7* gene in *Diabrotica virgifera virgifera*, significantly reducing pest survival [101]. These plants deliver dsRNA directly through pest feeding. Exogenous dsRNA formulations can be applied via spraying, root irrigation, or trunk injection. Following absorption by plants, dsRNA translocates through the vascular system to leaves/stems, triggering gene silencing upon pest ingestion [102]. This approach utilizes environmentally biodegradable dsRNA molecules.

Similarly, for *H. vitessoides*, dsRNA-based pesticides targeting essential genes can be designed for spray application or transgenic plant delivery to specifically disrupt pest development. Furthermore, research on chemosensory receptor genes (e.g., *HvitOR42*, *HvitOR43*) enables developing mating disruptors that impair chemical communication, thereby suppressing population spread. However, RNAi technology faces safety concerns and potential ecological risks, primarily regarding non-target effects, environmental persistence, and long-term ecological consequences. Theoretically, RNAi’s sequence-specific gene silencing minimizes impacts on non-target organisms [100]. Adopting bioinformatic tools (e.g., dsCheck) to optimize dsRNA design is critical for mitigating off-target effects [103]. Additionally, unmodified dsRNA is susceptible to environmental degradation via microbial enzymatic breakdown or UV damage. For instance, Dubelman et al. [104] observed soil half-lives of bare dsRNA as brief as 15–28 h, with rainwater washout accelerating dissipation. This rapid degradation mitigates long-term residual risks. While RNAi technology offers high specificity and low residue advantages for pest control, its ecological risks require comprehensive management through target optimization, formulation assessment, and resistance monitoring. Current regulatory frameworks (e.g., OECD guidelines) provide scientific foundations. Future *H. vitessoides* RNAi research should validate multigene co-regulation feasibility, enhance field-application stability/safety, and quantify long-term ecological impacts to balance agricultural sustainability with biodiversity conservation.

Genomic sequencing of *H. vitessoides* has revealed the molecular basis of its environmental adaptability. Key functional genes associated with detoxification, temperature stress responses, metamorphosis, and chemosensory signaling underpin *H. vitessoides* environmental adaptability and biocontrol potential. However, critical limitations and challenges persist: (1) Insufficient functional validation: Most studies rely on RNAi, with CRISPR/Cas9-based gene editing yet to be applied. (2) Multi-gene synergistic mechanisms: Temperature adaptation and detoxification likely involve gene networks, but current research focuses on single genes. Genomic data can be leveraged to construct a critical gene regulatory network governing the growth and development of *H. vitessoides*. (3) Limited application translation: Despite identifying potential targets (e.g., pheromone receptors), gene-driven or RNA-based control technologies have not been tested in the field.

**Table 3 insects-16-00690-t003:** Functional genes in *Heortia vitessoides*.

	Gene Name	Gene Coding Protein	Subcellular Localization	Developmental Stage Expression Pattern	Organizational Expression Pattern	Regulatory Factor	RNAi	Gene Function	Reference
Detoxification-related genes	*HvGSTs1*	Glutathione S-transferases	Cytoplasm (52.2%)	Expressed at all developmental stages, highest expression at pupal stage	In fourth-instar larvae, fat body expression significantly exceeded midgut, head, and cuticle expression; in 2-day-old adults, abdominal and thoracic expression significantly surpassed head, leg, and wing expression			Protects against toxic substances in the body	[66]
Temperature stress-related genes	*Tpx*	Thioredoxin peroxidase	Cytoplasm (69.6%)			*HvTpx* expression was induced by 0 °C, 10 °C, and 35 °C temperatures		Coping with temperature stress	[75]
	*CAT*	Catalase		Expressed at all developmental stages, highest expression in fifth-instar larvae	Expression in the fat body of fifth-instar larvae was significantly higher than in the head, cuticle, midgut, and Malpighian tubule; expression in the adult abdomen was significantly higher than in the head, thorax, legs, and wings	Expression of this gene was induced under high-temperature stress conditions (35, 37, and 39 °C)		Participation in resistance to heat	[76]
	*HvGP*	Glycogen phosphorylase	Cytoplasm	Expressed at all developmental stages, highest expression at egg stage	Expression in the fat body of fourth to fifth-instar larvae was significantly higher than in the head, legs, cuticle, midgut, and hindgut; expression in the adult wing was significantly higher than in the thorax, abdomen, head, legs, and antennae	A cold stress interval of 5–20 °C induced the expression of this gene; a heat stress temperature interval of 30–40 °C repressed the expression of this gene		Participating in the fight against hypothermia	[80]
	*HvAK*	Arginine kinase	Cytoplasm	Expressed at all developmental stages, highest expression in fifth-instar larvae	Expression in the head of fourth-instar larvae was significantly higher than in the fat body, midgut, Malpighian tubule, and legs	Both high- (35 °C) and low-temperature (4 °C) stresses caused the upregulation of expression		Responding to adverse environments	[82]
Metamorphic development-related genes	*HvTPS*	Trehalose-6-phosphate synthase		Expressed at all developmental stages, with the highest expression after pupation and before emergence	Expression in the fat body of fifth-instar larvae was significantly higher than in the head, cuticle, midgut, and Malpighian tubule		Both pupal and adult stages showed deformities, and survival significantly reduced compared to the control	Involved in biosynthesis of chitin and lipids	[88]
	*HvCDA1*, *HvCDA2*	Chitin deacetylase					*Heortia vitessoides* showed abnormalities or even death in the molting, pupating, and emergence stages, and all adults fortunate enough to have life characteristics showed wing folds	Requirements for the growth and development of *Heortia vitessoides*	[89]
	*HvEcR*	Ecdysone receptor	Mitochondria	Expressed in all developmental stages, with significantly higher expression in fifth-instar larvae and adults	Significantly higher expression in the fat body of fourth to fifth-instar larvae than in cuticle, head, midgut, and Malpighian tubule	Expression levels may be regulated by 20-hydroxyecdysone			[90]
	*HvFABP*	Fatty acid binding protein		Significantly higher expression from prepupal to adult stage than larval stage	Expression in the midgut of fifth-instar larvae was significantly higher than in the head, cuticle, fat body, foregut, and hindgut; expression in the wings of adults was significantly higher than in the head, thorax, abdomen, and legs	Starvation induces *HvFABP* expression, and 20-E induces upregulation of its expression	*Heortia vitessoides* dies because of molting failure	Participating in *Heortia vitessoides* molting process	[91]
Chemosensitive genes	*HvitOR42*, *HvitOR43*	Pheromone receptor			Highly expressed in the antennae of adult males (more than 10-fold greater expression than females)			May be associated with the recognition of female sex pheromones	[94]
	*HvitOR20*	Odorant receptor			Highly expressed in the antennae of adult females (more than 10-fold greater expression than males)			May play a role in female oviposition-related behaviors	[94]
	*HvitGOBP2*, *Hvit-PBP1*, *HvitPBP2*, *HvitOBP2*, *HvitOBP10*, *HvitOBP11*, *HvitOBP13*, *HvitOBP15*	Odorant binding protein			Highly expressed in the mouthparts of adult males (expression more than 10-fold greater than females)			May be associated with the recognition of sex pheromones	[94]
	*HvitCSP8*, *HvitCSP15*, *HvitCSP17*	Chemosensory protein			Preferentially expressed in males (more than 3-fold greater expression than females)			May be associated with the recognition of sex pheromones	[94]

Gene research provides novel insights for targeted control strategies. However, the outbreak dynamics of *H. vitessoides* remain intricately linked to the defense responses of its host plant, *A. sinensis*. Their interaction involves not only chemical signaling but also establishes a dynamic equilibrium through a tritrophic system (*H. vitessoides*–*A. sinensis*—natural enemies). The following sections will provide a systematic analysis of the molecular and ecological mechanisms underlying this interaction network.

## 5. Interaction Between *Heortia vitessoides* and *Aquilaria sinensis*

The interactions between herbivorous insects and their host plants constitute a critical component of terrestrial food webs. Over long-term coevolutionary processes, plants and herbivorous insects have developed intricate relationships [105]. The interaction between the oligophagous leaf-chewing pest *H. vitessoides* and its host plant *A. sinensis* exemplifies tritrophic dynamics, involving the pest, the host plant, and its natural enemy *Cantheconidea concinna*, a member of the family Pentatomidae (Figure 2). Understanding the mechanisms underlying these interactions will enhance our knowledge of plant–insect coevolution and provide a theoretical foundation for developing novel pest control strategies.

The tritrophic system involving *H. vitessoides*, *A. sinensis*, and *C. concinna* exhibits fascinating dynamics (Figure 2). Under natural conditions, healthy and undamaged leaves and flowers of *A. sinensis* release volatile organic compounds (VOCs) that attract adult *H. vitessoides* [66,67]. Four compounds—hexanal, (Z)-3-hexenyl acetate, nonanal, and decanal—extracted from the young leaves of *A. sinensis* have been identified as oviposition attractants for female *H. vitessoides* [106]. In flowers, 11 VOCs eliciting electroantennographic (EAG) responses in both male and female moths have been identified. Among these, heptanal and hexanal elicit the strongest EAG responses in females and males, respectively, with females showing a preference for benzaldehyde and males showing a preference for caryophyllene. Notably, *A. sinensis* flowers emit stronger floral scents at night, coinciding with the peak activity periods of *H. vitessoides* emergence, foraging, and mating, highlighting the pest’s adaptation to its host plant [107]. Following herbivory, *A. sinensis* can produce specific volatiles that repel *H. vitessoides* adults while attracting the predatory stink bug *C. concinna*. Damaged plants are more attractive to *C. concinna* than undamaged ones, with this attraction persisting for up to three days [108]. Beyond VOC-mediated interactions, *A. sinensis* employs direct defensive mechanisms against *H. vitessoides* feeding. Resistant *A. sinensis* plants exhibit strong inhibitory effects on larval feeding activity, impairing larval development. Specifically, neonate and early fifth-instar larvae avoid resistant plants, and larvae feeding on resistant leaves experience prolonged developmental periods, reduced survival rates, lower pupal weights, and shorter adult lifespans [109]. Further investigations into resistance mechanisms have revealed that the soluble sugar and protein contents in *A. sinensis* leaves are negatively correlated with resistance, while the flavonoid and tannin contents are positively correlated [110,111]. Our recent study explored the mechanisms underlying *A. sinensis* responses to *H. vitessoides* feeding and mechanical damage. The results indicate that the jasmonate (JA), salicylic acid (SA), and ethylene (ET) pathways are involved in the plant’s defense responses. Repeated mechanical damage enhances *A. sinensis* resistance to *H. vitessoides* larvae, with mature leaves potentially exhibiting stronger defensive responses than young leaves. The JA pathway plays a dominant role in this process [112]. Following mechanical damage (MD) and herbivore-induced wounding (HW), spatial compartmentalization of jasmonate biosynthesis gene expression was observed across different leaf tissues. Transcriptional analysis revealed that *AsOPR* (12-oxophytodienoate reductase, involved in JA precursor synthesis), *AsJAR* (jasmonoyl-L-isoleucine synthetase, responsible for JA-Ile activation), and *AsJATs* (jasmonate transporter) exhibited significant upregulation (*p* < 0.01) in mature leaves (ML). In contrast, their upstream biosynthetic genes—including lipoxygenase (*LOX*), allene oxide synthase (*AOS*), and allene oxide cyclase (*AOC*)—showed predominant activation in directly damaged younger leaves (YL) (Figure 2). This spatial distribution pattern suggests two potential mechanisms for JA-Ile accumulation in ML: (1) transport of JA-Ile or its precursors from locally damaged YL through the vascular system, or (2) de novo synthesis from mobile intermediates.

This tritrophic interaction system not only elucidates the coevolutionary dynamics between plants and herbivorous insects but also offers insights into the development of sustainable pest management strategies. However, notable limitations and challenges persist: (1) JA biosynthesis and downstream signaling: The compartmentalized JA biosynthesis mechanism (e.g., LOX-AOS-AOC in young leaves, OPR-JAR in mature leaves) requires further validation, particularly regarding signal molecule transport. Concurrently, the downstream transcriptional regulatory mechanisms remain uncharacterized. (2) Lag in resistant cultivar development: Although resistant *A. sinensis* varieties (e.g., high tannin content) have been identified, their genetic basis remains uncharacterized, limiting molecular breeding applications. (3) Ecological effects of tritrophic interactions: The stability of natural enemy and plant defense synergies in complex field environments needs evaluation.

## 6. Conclusions and Outlook

*H. vitessoides*, a major defoliating pest of *A. sinensis*, poses a severe threat to the sustainability of agarwood plantations due to its outbreak potential. This review focuses on summarizing the molecular drivers underlying *H. vitessoides* damage. Key genes governing its growth and development play crucial roles in detoxification, temperature stress response, and metamorphosis, enabling environmental adaptation via the antioxidant enzyme system, energy metabolism regulation, and chemosensory gene networks. Concurrently, *H. vitessoides* interacts with its host within a tritrophic defense system (*H. vitessoides*—*A. sinensis*—natural enemy *Cantheconidea concinna*) mediated by the host’s jasmonic acid (JA) signaling pathway and emission of volatile organic compounds (VOCs). Although existing green control technologies (e.g., pheromone trapping, combined application of *Beauveria bassiana* and chemical pesticides) show promise, they face challenges such as insufficient field stability and lack of ecological risk assessment. Future research should prioritize the following key areas:Gene Editing and Functional Validation: Utilize CRISPR/Cas9 technology to target and knockout key genes in *H. vitessoides*, elucidating the regulatory networks governing its development and behavior.Field Application of RNAi Technology: Design double-stranded RNA (dsRNA) formulations targeting multiple genes (e.g., detoxification enzymes, chemosensory genes) based on current research for environmentally friendly field deployment.Constructing Cross-Species Molecular Interaction Networks: Employ multi-omics integration to decipher the molecular mechanisms underlying natural enemy responses to host signals. Systematically analyze the induced expression patterns of *H. vitessoides* genes in response to host defense compounds, while identifying the temporal regulation of JA pathway genes in *A. sinensis* triggered by herbivory. This will reveal the dynamic molecular interaction network of “host defense—*H. vitessoides* response”.Multi-Omics-Driven Integrated Modeling: Combine genomic, transcriptomic, and metabolomic data to construct multi-scale models of *H. vitessoides* environmental adaptability and population dynamics. Develop time-delay differential equation models to simulate the nonlinear regulatory effects of different control measures (e.g., RNAi spray frequency, natural enemy release intervals) on population density, optimizing control strategies.

In summary, future research aims to establish a precision control technology system integrating “intrinsic molecular mechanisms—extrinsic environmental regulation”, providing a theoretical basis for targeted intervention of *H. vitessoides* outbreaks.

## Figures and Tables

**Figure 1 insects-16-00690-f001:**
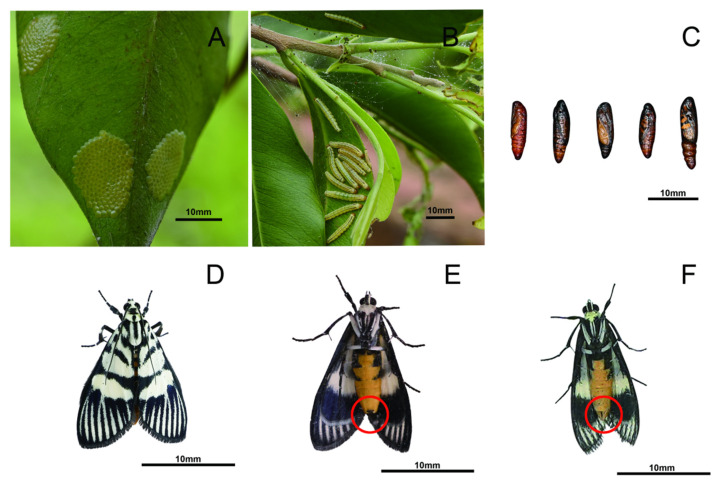
Morphological diagram of *Heortia vitessoides* in different development stages. (**A**) Egg; (**B**) larva; (**C**) pupa; (**D**) dorsal side of adult; (**E**) ventral side of female adult, showing a stout, orange-yellow abdomen with blunt ends, and the red circle indicates short black tufts around the genital hole; (**F**) ventral side of male adult, showing an elongated, yellow abdomen with pointed ends and clusters of long yellow hairs, with the red circle indicating the absence of short black tufts around the genital hole.

**Figure 2 insects-16-00690-f002:**
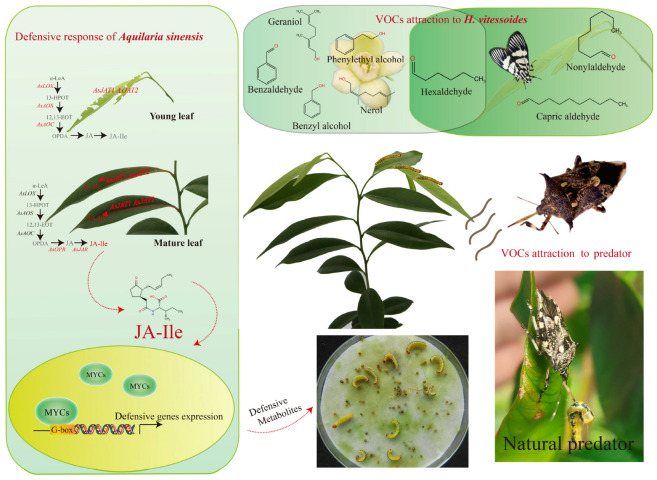
Interaction mechanism of the three-tier trophic system (edited by Chen et al., 2023 [71]). α-LeA, linolenic acid; 13-HPOT, 13-hydroxyperoxy-octadecadi(tri)enoic acid; 12,13-EOT, 12,13(S)-epoxy-octadecaenoic acid; OPDA (12-oxophytodienoic acid); OPR, 12-oxo-phytodienoic acid reductase; JA, jasmonate; JAR: jasmonyl isoleucine conjugate synthase; JA-Ile, jasmonyl-isoleucine; JAT, jasmonate transporter.

**Table 1 insects-16-00690-t001:** Occurrence regularity of *H. vitessoides* in different areas.

Region	Annual Generation Algebra	Damage Period (Month)	Damage Peak Period	Monthly Average Temperature in Peak Period (°C)	Monthly Average Humidity During the Peak Period (%)	Average Monthly Rainfall During the Peak Period (mm)	Annual Average Temperature (°C)	Annual Average Humidity (%)	Mean Annual Precipitation (mm)	Reference
Hainan, China	8~10	2~11	April	24.9~29.1	73~85	30.3~124.7	22.5~25.6	81~83	1500~2500	[4,21]
Guangdong, China	6~8	4~12	-	-	-	-	22~22.6	74~82	2040~2888.1	[5]
Guangxi, China	5~6	3~12	Late April to early May; October	17.7~24.6; 18.1~22.3	76~83; 66~83	80.8~351.9; 53.6~102.7	20.7~22.27	75~83	1086.8~1569.3	[22]
Yunnan, China	6	-	April to May	5.5~22.5	56~80	18.5~194.4	22.6	74~77	1136.6	[23]
India	4~5	2~9	-	-	-	-	24~27	72.9	1200	[24]
Indonesia	-	All year around	July to September	-	-	-	23~32	77~85	3950	[12]

**Table 2 insects-16-00690-t002:** Common insecticides for *Heortia vitessoides*.

Drug Type	Drug Name	Insecticidal Efficiency (%)	Concentration	Insecticidal Principle	References
Antibiotics	Emamectin benzoate	100.00	5.0 × 10^6^ dilution of 0.5%	Contact killing and stomach toxicity	[36]
	Avermectins	100.00	5.0 × 10^6^ dilution of 1.8%	Contact killing and stomach toxicity	[36]
Mixed class	Sendebao	98.90	30 times of synergistic powder		[38]
Insect growth regulators	Fenoxycarb	98.90	8000 dilution of 3%	Contact killing, stomach toxicity, and exhibiting strong juvenile hormone activity	[38]
	Spinetoram·methoxyfenozide	100.00	Diluted 1000-fold		[11]
Plant source	Eucalyptol SL	90.88	1000 times of 5%	Mainly contact killing	[39]
	Matrine		0.30%	Paralysis	[40]
Microbial source	Spinetoram	100.00	Diluted 1000-fold	Contact killing, stomach toxicity, and interfering with nerve activity	[11]
	*Beauveria bassiana*	84.00~90.00	2.4 × 10^8^ spores/mL	Infection	[24]
	*Metarhizium anisopliae*	40.00~52.00	2.4 × 10^10^ spores/mL	Infection	[24,41]
		100.00	1 × 10^9^ spores/mL		
	*Aspergillus nomius* Q527	83.30		Infection	[42]
	*Helicoverpa. armigera* NPV		2 billion PIB/mL	Infection	[40]

## Data Availability

The original contributions presented in this study are included in the article. Further inquiries can be directed to the corresponding authors.

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
