# Peer review of "Heortia vitessoides Infests Aquilaria sinensis: A Systematic Review of Climate Drivers, Management Strategies, and Molecular Mechanisms"

_insects, 2025, doi:10.3390/insects16070690_

Round 1

Reviewer 1 Report

Comments and Suggestions for Authors

The manuscript considers numerous aspects of the ecology and genetics of Heortia vitessoides also in relation to Aquilaria sinensis.

In my opinion some parts of the manuscript are not strictly necessary and could be reduced, however, the paper is very articulate and fulfills the specific role of review with an extensive bibliographic collection. A perplexity of mine remains referring to the number of authors and co-authors signing the paper (there are as many as 12 belonging to 3 different institutions and research centers), which is quite unusual for a review paper. The number of self-citations is also very high. Overall, the manuscript is more appropriate as a book chapter and not as a contribution of research activity. Its acceptance on the scientific Journal Insects, must be decided by the Editor. In any case title of the paper must be changed.

Reviewer 2 Report

Comments and Suggestions for Authors

As a reader without prior knowledge of the system, the review manuscript titled “Genetic Insights and Integrated Control Strategies against Heortia vitessoides: Safeguarding Aquilaria sinensis Cultivation” seems like a relatively comprehensive review of work that has been done on the pest H. vitessoides, and serves to collate existing knowledge in the system as well as some areas of future research related to its management. I am however not sure how much value this paper has outside of the specific system. There seems to be relatively few new or more widely generalizable ideas emerging from this review, rendering the scientific knowledge gain of this review limited, although the species itself is widespread and of economic importance. However, I believe the wider value to the scientific community of this manuscript can be improved on by bringing forward the main conclusions and areas lacking research in the Conclusions and outlook section 7 more clearly, as these ideas can be applicable across diverse systems. This could be accomplished by having subtitles in the section 7, and organising and highlighting the main conclusions and aspects identified in need of future research for example as separate text boxes or list of questions.

Overall, the manuscript is well-written and easy to follow. In addition to the new knowledge gained from the review I was hoping to see more discussion on the ecological aspects of the system. For example, although climate factors were identified as an essential topic of future research, little research was mentioned about how changes in local climatic conditions affect H. vitessoides abundance or population demographics in the field, and if there are any interactions of this with e.g. resource availability, predator abundance, population density or season.

I was also waiting for more discussion on the potentially important ecological interactions and dynamics in relation to possible management actions. What kind of ecological data would be needed for population dynamic modelling? It would also be nice to have dynamic models to test the efficacy of different control strategies to support decision-making between alternative approaches. For example, even though the management strategy of catching adult moths seems efficient by numbers and can probably reduce the bulk of eggs laid, it is likely not enough to stop a new generation from forming (given that only a few moth individuals have to succesfully mate and lay eggs to produce huge numbers of offspring). However, if the larvae grow slower in lower densities, reducing the number of reproducing adults might still be an effective and strategy. If combined with pheromone-trapping or other chemical lures to make the traps more specific to this pest species, this could also be a biodiversity-friendly approach.

The ecological consequences of the suggested management strategies could also be discussed in more detail. I am not sure I understood how knowledge of genes involved in climate adaptation translates to pest control in this system. Does it help with more accurate modelling? Or if the main application is to target these important genes to silence them, what is the next step then? If genetically modified individuals are not fit, how will releasing them help fight an existing or future outbreak? If you were to release genetically modified individuals, at what life-stage would that be? Could sexually mature individuals carrying deleterious genes be released to mate with the wild population to reduce their viability? Is it safe to release reproducing genetically modified organisms?

On a similar note, is it safe to release infectious microbes and generalist predatory insects at an ecologically sensitive area? How will these spread to the environment and which other species will they affect?

Minor comments:

Page 2 paragraph 2: The sentence describing the economic and social importance of agar wood is a little vaque and does not seem to provide much information – medicine use was already mentioned and it is unclear what the daily chemical products or various other industries are? I suggest giving examples of at least some of those other industries or uses.

Page 2 Section 2.1. first sentence: Can remove the second verb “are” as there is already a verb in the sentence. I suggest “The eggs of H. vitessoides are oblate, densely arranged in scales, and measure 0.5 to 0.8 mm in diameter (Figure 1A).”

Page 2 Section 2.1. third sentence: Any reference for the hatching success rate? Has this been measured in the laboratory or in the field and does it vary geographically or temporally?

Figure 1 legend. Please use the terms dorsal and ventral instead of “reverse” and ventral.

From Figure 1 it seems that the moth possesses aposematic coloration (high contrast of black/white/yellow colour). Do you know if it is also chemically defended? Are there any avian predators attacking it, or are the stink bugs the only known predators? If they are indeed aposematic, aggregation may also increase their defence efficacy by increasing signal size. The larvae seem cryptic in coloration, but are they chemically defending such as e.g. pine sawfly (Diprion pini) larvae?

Section 2.2. second sentence. Any reference for the mortality when separated from their group? Is this really because they choose not to eat, or do they need the group to be able to produce enough chemicals to dissolve the leaf surface and eat it? How was this observation made, could the larvae have been damaged in the separation process leading to subsequent mortality?

In the same paragraph, you refer to “recent research” without a citation. Please add the relevant citation here.

Section 3.1 The “and other regions” in the end of the list of countries and regions is a bit vaque. Could instead use “other (sub)tropical regions”, or something to delineate a little more accurately?

This whole section 3.1 also repeats the information in Table 1. Could the information of outbreak timing by region be arranged by latitude or some other relevant variable varying between the sites?

Section 3.2 the effective accumulated temperature rule seems to be of high importance, and its usage in modelling could be mentioned again in discussion.

Section 3.2 European corn borer needs to be mentioned by scientific name as all other species in the manuscript.

Section 5. The sentence “This review focuses on the tritrophic interactions among the oligophagous leaf-chewing pest H. vitessoides, its host plant A. sinensis, and its natural enemy Cantheconidea concinna (Figure 2).” Is surprising, as no mention of C. concinna exists before this section. Please reword to match the rest of the manuscript.

Section 7. Is drought relevant in this context, are the A. sinensis plantations not watered?

Reviewer 3 Report

Comments and Suggestions for Authors

For a Review paper, I am left asking some critical questions after reading the manuscript.  

  1. In the Introduction/Background, I would expect the authors to carefully point out the gaps in knowledge that they have found and will address in this review.  In other words, why is this review needed?  
  2. The authors state in the Introduction that the literature and practice of control of Heortia is not integrated across biological and genetic domains, so presumably the review paper is written to bridge the gaps and provide integration.  However, it does not.  The paper is structured with exactly the same domains and builds very little connection between them. In this sense the Review paper does not accomplish its goals. 

In addition to these overarching problems with the design and organization of the paper, there are some more specific  aspects that need attention.

Since no line numbers are provided in the manuscript, it's difficult to give exact locations for the comments below.  

In the Abstract, give family and order as in the Intro, and family for the host plant.

Table 1 - don't start at the bottom of the page.

Section 3.2. what is the "effective accumulated temperature"?

Section 4.  It is not clear how much of the genetic information present is common to a wide range of insects, or even all insects, and what is specific to Heortia vitessoides.  How much of this literature needs to be repeated here?

Section 5. As before, give family of natural enemy C. concinna

Round 2

Reviewer 2 Report

Comments and Suggestions for Authors

I thank the authors for their responses to my questions and concerns, and I am content with most of the answers. However, the changes/edits made in the manuscript need to be checked, especially for "Conclusions and outlook"-section. Please find more detailed comments below.

There is a typo in the title, Mangerment should be Management.

Could you mention in the abstract or beginning of intro that we are talking about a moth and a tree, e.g. the English names for the two species?

L62-64 the second added sentence is missing a verb, such as "is".

I like the addition of geographic description in the intro.

L105-107 The sentence "Notably, H. vitessoides exhibits aggregation and feeding behaviors, with its growth and development being influenced by group size." could be rewritten for linguistic reasons. How about "Notably, H. vitessoides growth and development are influenced by group size:"

L108 Is it really food refusal, or rather that they are not able to forage/digest the leaf when in low numbers? The reference you cite says "that gregariousness may play a critical role in facilitating feeding for newly hatched larvae". Please rephrase not to speculate on the causal mechanism of death in isolation.

L656 the subtitle number is off, should be 5.1, not .1

L679 again, the subtitle number is odd. 

L680 I assume this paragraph should not be bolded and set as a title? This paragraph also has some extra brackets and words that do not seem to belong, and is cut short. Is something missing completely?

L690 Conclusions and outlook should be a subtitle. Please check the whole section, parts that should not be bolded are bolded.

L725-734 This whole paragraph about genetics does not seem to belong here, or at least was not included in the response to reviewer for this section?

Please fix the reference list, numbers are repeated and mixed at least on lines 813, 815 and 992 onwards.

Comments on the Quality of English Language

English language is understandable and clear for the most part, but some sentences are very long and could be shortened for clarity. A few sentences are missing a verb. Focal species names are not mentioned in English.

Reviewer 3 Report

Comments and Suggestions for Authors

The paper has been improved.   However, clarification is still needed for Effective Accumulated Temperature on lines 179-183.  As stated, a simple sum of daily temperatures above a developmental threshold does not produce anything meaningful.  Is this simply another way of referring to degree-day calculations?  If so, then just explain it and cite an appropriate reference. 

There are several instances where Chinese characters randomly appear in the text, e.g line 597.  Copy editing is needed.
